# "I Will Show You My Faith by My Works": Addressing the Nexus between Philosophical Theodicy and Human Suffering and Loss in Contexts of 'Natural' Disaster

**Roger Philip Abbott**

The Faraday Institute for Science & Religion, Cambridge Theological Federation, The Woolf Building, Madingley Road, Cambridge CB3 0U, UK; rpa24@cam.ac.uk

**Abstract:** As a practical theologian and researcher in the field of 'natural' disasters, as well as being a disaster responder chaplain, I am often confronted by, and have to confront, the nexus between theology/philosophy and "real life" in extremely traumatic contexts. The extreme suffering that is often the consequence of catastrophic natural disasters warrants solutions that can help vulnerable populations recover and adapt to live safely with natural hazards. For many practice-based responders, speculative theological/philosophical reflections around situations that are often human-caused seem predominantly vacuous exercises, potentially diverting attention away from the empiricism of causal human agency. In this article, I explore a middle ground involving a nuanced methodological approach to theodicy that is practical but no less intellectually demanding, that is theological more than philosophical, practical more than theoretical; a middle ground that also takes seriously the human culpability as causal for the human, and some would say the divine, suffering from disasters. I will include in this exploration my ethnographic fieldwork following the catastrophic earthquake to hit the Caribbean nation of Haiti in 2010.

**Keywords:** disaster; suffering; evil; practical theology; philosophical theodicy; practical theodicy; anthropodicy; social justice

## 1. Introduction

The opening words in the title of this paper, taken from the Epistle of James 2:18 (ESV 2002), are part of a larger statement where the author, by constructing an imaginary conversation between two persons, contrasts a theoretical faith with a faith that works, a faith where the beliefs lead to concomitant actions. The person uttering the words in my title is meaning, "If you want to know my faith, read my works; if you want to know the basis for my works, read my faith [read, theology]."

The results of my exposure to, and research into, 'natural disasters' has seriously challenged my use of the term *natural* disasters when there has been much more empirical evidence of human agency that has turned a natural *hazard* into a disaster.[1,2] The term 'natural disaster,' at least when used in a

---

[1] Since 1989, I have been a responder to major emergencies/disasters as a Christian chaplain. Since 2012, the role of responder has been added to my role as Senior Research Associate in Natural Disasters, The Faraday Institute for Science & Religion, Cambridge, UK. I have carried out research projects after Hurricane Katrina, which occurred in 2005, the Haiti earthquake (2010), the Super-Typhoon Yolanda (Haiyan) in the Philippines (2013), and on the Somerset Levels flooding (2014).

[2] According to the International Disaster Data-base (EM-DAT), natural hazards are categorised under the following: geophysical; meteorological; hydrological; climatological; biological. Available online: https://www.emdat.be/classification (accessed on 10 October 2018).

religious context, tends almost always to focus upon whether there has been some metaphysical or divine causation and culpability. It is generally recognized that the academic philosophical pursuit of a theodicy emerged as being deeply problematic following the Lisbon earthquake, tsunami, and fire of 1755. The terrible human toll from this disaster cast serious questions over the predominant philosophic theodicy of that era, namely that of the Leibniz (1710) theodicy of 'The best of all possible worlds.' Or, what is, is for the best (Voltaire 2006; Voltaire 1977). From Lisbon through the modern era, different attempts at philosophical theodicy have been pursued, as Kenneth Surin's summary account explicates (Surin 1993, pp. 193–95). Most recent notable theodicies are those according to Plantinga (1974) (free will defence), Swinburne (1979) (natural law), Hick (1985) ('soul making'). These have been largely philosophical theories. Theologians such as Southgate (2008) (suffering intrinsic to the evolutionary process), Fretheim (2003) and Moltman (1974) (a suffering God), have offered their more theologically theodic versions. Though primarily a biologist, Denis Alexander has added his theological reflection (Alexander 2008, pp. 277–92) to address the issues of natural evil and suffering.

This focus maintains an academic obsession for solving the mystery of suffering and evil while distracting from the more obvious and immediate human causation, which, if addressed constructively and responsibly, could relieve large amounts of suffering and losses when natural hazards occur. This focus also diverts attention away from the empirical evidence for the benefits accrued from adaptation, and cultural transformation (Schipper 2015, p. 146) to natural hazards. However, as an academic theologian, I am constantly challenged at the nexus between speculative theology/philosophy, which have played a part in the formation of my theological convictions and practices, and the traumatic sequelae that casualties of these kinds of disasters have to recover from, and which pastoral responders are called upon to address.

On the one hand, as an academic theologian, I am used to the discourses on the metaphysical problem of God, evil and suffering, and on theodicy, and finding no satisfactory philosophical solution. Such discourses frequently follow in the wake of some catastrophic event, and the focus is inexorably, so it seems, upon God: why *God* allows terrible things to happen even to good people, why *God* cannot or will not prevent suffering, why *God* allows nature to become so wildly out of control (Alexander 2008, pp. 277–92; Hart 2005, pp. 7–15). This perspective has been dubbed the "God's eye view" (Griffioen 2018). Voltaire gave voice, in his passionate critique of Leibniz (1710) *Essais de théodicée*, following the Lisbon earthquake, tsunami, and fire on 1 November 1755, to the luxury that armchair philosophical and theological theorists enjoy following such catastrophes. He wrote (Voltaire 1977, p. 15),

> O tranquil minds who contemplate the pain
> And ship-wreck of your brothers' battered forms,
> And, housed in peace, debate the cause of storms,
> When once you feel Fate's catalogue of woe.

Voltaire could not untie the "strange knot" that the Lisbon earthquake tightened for him (Voltaire 1977, p. 19), and he was certain the philosopher Leibnitz had not done so either.

My concern is that, after so many centuries of trying, there is no philosophical theodicy that has succeeded in untying Voltaire's theodic "knot." Yet, in academe it seems, all too often, that untying this "knot" persists as the philosophical exercise that counts most for academic credibility even though God and evil remain unresolved, and human suffering remains untouched when the focus is on a speculative philosophical 'God's eye' view of untying the "knot."

As an academic ethnographic researcher of disasters, for an Institute that specialises in a partnership of science and religion, I can see, more clearly than ever I have, the empirically substantiated problem *humans* are when it comes to 'natural' disasters. I see how much suffering could be spared, and how many lives could be saved, if *humans* took their divinely mandated, moral responsibilities to each other (and to God) seriously. Consequent policies, attitudes, and lifestyles

could alleviate so much suffering and prevent deaths, life-changing injuries, and livelihood losses. The benefits could be so much more immediate than any gained through repeating the centuries-old speculative metaphysical discourses that have made little discernible practical difference to disaster risk reduction since "the first 'modern' disaster" in Lisbon in 1755, when, in Voltaire's words in a letter to a friend, Leibnitz' philosophical theodicy "got it in the neck." (Voltaire 2006; Dynes in (Braun and Radner 2005); Leibniz 1710).

How can the philosophical and the practical be resolved most effectively methodologically is the question this article is addressing. In this article, I explore a middle ground, involving a nuanced methodological approach to theodicy that is practical but no less intellectually demanding, that is theological more than philosophical, practical more than theoretical. It is a middle ground that also takes seriously the human culpability as causal for the human, and some would say, the divine suffering from disasters (Hall 1986; Fretheim 2003; Moltman 1974). I will case-study this exploration using my ethnographic fieldwork following the catastrophic earthquake to hit the Caribbean nation of Haiti in 2010.

## 2. The Problem

The polarity between the philosophical theodicy and the alternative practical theodicy preferred, for example, by Forsyth (1916); Hall (1986); and Swinton (2007) is exposed in the caveats prefacing the theodical reflections of certain contemporary philosopher theologians. For example, Plantinga (1974, p. 29) states quite explicitly that,

> Neither a Free Will Defense nor a Free Will Theodicy is designed to be of much help or comfort to one suffering from such a storm in the soul . . . Neither is it to be thought first of all as a means of pastoral counselling. Probably neither will enable someone to find peace with himself and with God in the face of the evil the world contains. But then, neither is intended for that purpose.

Don Carson admits his book, *How Long, O Lord?* is "not even the sort of book I would give to many people who are suffering inconsolable grief." (Carson 1999, p. 9). In the light of such caveats, we are entitled to ask what value such works could possess that is more than merely speculative? What transformative or redemptive value do they contribute to catastrophic contexts of suffering and grief? What compassion to the sufferer do they intend? Hence, Surin (1993), concludes of philosophical theodicy,

> Theodicy, it could be said, is always doomed to be at variance with the profound truth that the "problem of evil" will cease to be such only when evil and suffering no longer exist on this earth. Until that time there is much substance to the charge that the theodicist's presumption . . . only trivializes the pain and suffering of those who are victims. It is therefore necessary to stress that we are not likely to bring much comfort to the victims of suffering with theodicy.

Rightly, therefore, in my view, even in his more practical apologetic work, *Doors of the Sea*, Hart (2005, p. 99) insists that, "... words we would not utter to ease another's grief we ought not to speak to satisfy our own sense of piety."

However, if Hart is right in his counsel regarding *words*, should this mean an end to theoretical, even speculative *thoughts* within the protected spaces in academe? Should there not be space where philosophical and theological theorising can take place without it being assumed to be a product for immediate pastoral application? Is not the current focus upon reflective practice, as an aspect of practical theology (Bennett et al. 2018; Schon 1991), actually grounded, at least in part, by philosophical and theological speculation from within academe? Should the quest for an answer to the question, 'Why does *God* allow natural disasters to happen?' continue, or should the focus turn to the more empirically driven question, 'Why do *humans* allow natural disasters to happen?' Could answering the latter question provide a major contribution to resolving the former?

The narratives presented by my research participants, the majority of whom self-identified as Christian, and all of whom had been exposed to catastrophic disasters, demanded answers from *humans* (institutions, politicians, non-governmental organisations, contractors, society) more than from God, even though participants held strong views on the sovereignty of divine providence and on practicing prayer, views that were immensely comforting to them pastorally. From a few hundred participants, less than a handful raised any desire to interrogate God or to hold God to account in a negative way for their plight. Interestingly, my anecdotal experience as a responder-chaplain to major incidents abroad and in the UK has also shown me very similar perspectives from survivors of catastrophic incidents.

My work in an academic research institution brings me into close proximity with speculative philosophical and theoretical theological discourses on the issue of theodicy. These discourses involve interrogations of God that I, as an academic spectator, could well imagine bereaved and survivors of such incidents asking, but which have rarely ever been asked of me, or raised in my hearing by survivor-sufferer participants, despite the catastrophic nature of the disasters they have experienced. My work as a researcher of so-called natural disasters globally, and even as a responder-practitioner to the more obvious human disasters in the UK, has not brought the philosophical/theoretical questions of theodicy to the fore from suffering survivors or from the bereaved. The responses survivor-sufferers and the bereaved have volunteered and the help they have sought, including the questions they have raised, and the solutions they have proffered, have been much more worshipful and prayerful, but also more practical, obvious, and achievable.

Having spent many months conducting fieldwork in contexts of seismic, meteorological, and hydrological hazards that have turned catastrophic, listening to the stories of survivors I realise that there are achievable solutions staring us in the face involving *human* factors that could transform hazardous situations into contexts of relative safety through human agency adaptation. So a part of me, even as a theologian, finds I am more at home with the pragmatic realism of anthropologists and social geographers, such as Oliver-Smith (2010) when he asserts,

> In short, disasters are not accidents or acts of God. They are deeply rooted in the social, economic, and environmental history of the societies where they occur. Moreover, disasters are far more than catastrophic events; they are processes that unfold through time, and their causes are deeply embedded in societal history. As such, disasters have historical roots, unfolding presents, and potential futures according to the forms of reconstruction. In effect, a disaster is made inevitable by the historically produced pattern of vulnerability, evidenced in the location, infrastructure, socio-political structure, production patterns, and ideology that characterizes a society.

In a similar vein, Smith (2006), commenting upon the aftermath of Hurricane Katrina said,

> It is generally accepted among environmental geographers that there is no such thing as a natural disaster. In every phase and aspect of a disaster—causes, vulnerability, preparedness, results and response, and reconstruction—the contours of disaster and the difference between who lives and who dies is to a greater or lesser extent a social calculus. Hurricane Katrina provides the most startling confirmation of that axiom. This is not simply an academic point but a practical one, and it has everything to do with how societies prepare for and absorb natural events and how they can or should reconstruct afterward. It is difficult, so soon on the heels of such an unnecessarily deadly disaster, to be discompassionate, but it is important in the heat of the moment to put social science to work as a counterweight to official attempts to relegate Katrina to the historical dustbin of inevitable "natural" disasters.

When the managing director of the International Monetary Fund, Dominique Strauss-Kahn, implied, following the 2010 earthquake, that Haitians needed to "escape their cycle of poverty and deprivation *fuelled by merciless natural disasters*," Chancy (2013, p. 200, emphasis mine) responded, tellingly,

Though laudable in intent, Strauss-Khan's remarks suggest that only natural disasters have had a hand in producing Haiti's cyclical poverty and also that the international community's response is one bound up in a response to what cannot be helped, that is, an act of God. Given the religious rhetoric that enveloped Haiti in the aftermath of the earthquake . . . I have to wonder why the international community's response is steeped in neoreligious ideals of pity or mercy rather than in redressing of political wrongs.

Ironically, as long as disasters involving natural hazards are regarded as natural, then, at least for the religious and the antireligious, the focus will be upon God, since humans have little control over the powers of nature/creation, and God, or the notion of God, is assumed to be the controlling force and, therefore, to blame for deaths and suffering. However, for me *as a Christian and theologian*, the factors identified in the above quotes raise a much overlooked issue in the philosophical/theological discourse, namely, why do *humans* cause suffering? In the words of the Jesuit scholar, Jon Sobrino, "In today's context, it has been easier to apply suspicion and critical judgment to God, whom we do not see, than to the reality that human beings have created, which we see very well." (Sobrino 2006). Addressing the reality of *human* evil could be much more constructive and productive for administering comfort, and hope, as well as for disaster mitigation, than speculating upon natural evil from which little comfort or mitigation result.

The extreme suffering that is often the consequence of catastrophic disasters requires discourses and *solutions* tied to reflective practices that are performative and transformative (Bennett et al. 2018; Swinton 2007; Graham et al. 2005), where, in theological terms, orthodoxy can be audited in orthopraxy (Anderson 2001). Or, in the Apostle James' words, where faith can be seen in works that transform future vulnerability into greater adaptability to both natural and human-made hazards (Smit et al. 1999; Adapting to Disasters Is Our Only Choice 2010). However, each practical strategy we come up with is certainly indebted in part to some kind of theoretical theological and philosophical reflection that has taken place at some time in the practitioner's past and in the research literature that has been formative to their evidence-based pastoral praxis.

The challenge facing academics and practitioners, therefore, regarding catastrophic disasters, is a methodological one more than a philosophical one. Can we form ways of ensuring that reflective metaphysics and theology collaborate with and contribute to the discipline of practical theology (Abbott 2013, pp. 33–40; Ballard and Pritchard 2006; Anderson 2001; Woodward and Pattison 2000), and to an approach where practical theology can establish the survivor-sufferers' ethnographic milieu in which such reflective theology can operate transformatively? Is there a methodology that can ensure that words we would not utter to ease another's grief we will not speak to satisfy our own sense of piety, to cite Hart's words?

It used to be relatively easy to engage in speculative philosophy and theology within the safe confines of the affluent Western Enlightenment-minded academe, insulated from the grinding poverty of many Low Income Countries where 'natural' disasters strike the hardest and people suffer most (Bankoff 2010). In contemporary academe where the demand for an 'empirical turn' places a need for practical, achievable outputs that validate the financial inputs from research funders, then the age-old pursuit of philosophical theodicy is being challenged, fiscally and conceptually.

## 3. Toward a Solution

Peter Hicks, after outlining the various theodicies, still concludes, "Many have found one or other of these suggestions helpful as they struggle to understand why God should allow evil and suffering in the world," and "it is helpful to know there are possible reasons; the agony of struggling with an insoluble problem is removed." (Hicks 2006, p. 151). However, I am not convinced that the problem is removed simply by choosing to believe a particular philosophical theodicy, which is but one among many, over which people have speculated indecisively for so long. Not one of my participants made reference to a philosophical theodicy, let alone to one being a comfort to them, though many appealed to a theology of divine providence as their psychological and spiritual life-line.

John Swinton adds his own voice in protest against philosophical theodicy by averring it is pastorally problematic and theologically questionable because it has the potential for becoming a source of evil more than a cure (Swinton 2007, p. 3). For Swinton, theodicy is a practical problem requiring a practical response if sustaining hope in God's providential love is to be achieved. However, while refusing to denounce intellectual reflection, he recommends, "we take seriously the contribution of intellectual activity in responding to the problem of evil but recognise that intellectual activity is not an end in itself, but rather a means for developing transformative perspectives and practices that will enable faithful living." (Swinton 2007, p. 70, see also pp. 17–29).

Accepting the strength of Swinton's point, I suggest that there is a role for philosophical theological reflection upon the issues of God, evil, and suffering when it is conducted at the appropriate time and in the appropriate environment, and when it is pursued from a pragmatic perspective.[3] In the realm of nursing training, partnerships between academe and the practice are being pursued constructively (Barger and Das 2004). The empirical sciences require a similar methodology. For example, before pharmaceutical drugs can be released into the market for public consumption, they require rigorous experimentation and testing in the safe and secure environment of the laboratory (Bearn 1981). Surely, a similar approach is legitimate for exploring and testing theological understandings of principles for pastoral care and recovery from catastrophic disasters? Is it not the role of academe to be a laboratory of ideas, where theoretical concepts can be discussed and tested before being assumed to be safe for public consumption? Is such a facility not essential for the protection of sufferers? Could it not be argued that such an intellectual 'laboratory' discussion between philosophers like Leibnitz, Voltaire, and Rousseau, provoked by the "first modern disaster," the Lisbon earthquake of 1 November 1755, was instrumental in developing the Portuguese government minister, the "paradox of the enlightenment," Marquis of Pombal's innovative model for emergency response? (Bankoff 2015, p. 60; Dynes, in (Braun and Radner 2005)). After Lisbon, the focus of attention moved from a predominantly religious to a more pragmatic and practical, and also socially scientific, perspective. (Dynes, in (Braun and Radner 2005); Dynes 1999). In fact, Dynes (1999, emphasis mine) maintains that, "the most profound effect the earthquake had on ideas was its consequences for certain intellectual currents that were already evident in other European capitals. Those intellectual currents, generally thought of as comprising the Enlightenment, are *now considered as the seed bed for political and social thought within the western world*."

The philosopher, Helm (2008, emphasis mine), while critiquing the speculative philosophical approach of Plantinga, averred,

> At one point in his book God, Freedom and Evil, Alvin Plantinga says that he is offering philosophical enlightenment in connection with the logical problem of evil and that he is leaving to others the pastoral problems arising from encountering evil. *But we have seen that the issues of philosophy, theology and the occurrence of personal evils in a life should not be so tidily boxed*. Part of a fully Christian philosophical response to evil involves identifying and rejecting the unbiblical and consequently sub-Christian conceptions of God that are rife in so many 'Christian' philosophical responses to it. *For Christians, philosophy and theology should not be separated, nor should philosophy and pastoral care*.

Helm's warning that some theological underpinning of pastoral care can be more informed by questionable and unbiblical concepts is a valid one, making such concepts unsafe to administer pastorally, and warranting academic challenge. As an example, from my own research, in every disaster context I have worked in, there were cases where the disaster was blamed upon the sins of individual people, or upon the specifically social and/or religious life of some group, by theologies that are highly questionable, exegetically and ethically, contra the foci of Jesus (Luke 13:1–5; John

---

3　I suggest that Carson's book (Carson 1999) fits this description admirably, despite my critique above.

9:2–3) (e.g., Robertson 2010). Therefore, this historically common faith perspective (Webster in (Braun and Radner 2005); Abbott 2013, pp. 127–28, n. 32) makes the role of academe as laboratory all the more important. Ruard Ganzevoort may have a valid point, therefore, when he believes that there is room for some theological-philosophical investigation into theodicies of trauma, though he agrees that regarding these as "psychological tools" more than as academic speculative theories for providing final answers is more helpful.

Theologically, the issues of evil and suffering are not shirked within the Judeo-Christian Scriptures, from both human and divine perspectives. Whether it is correct to understand any Biblical accounts as theodicies is an extremely moot point. Miroslav Volf claims that the perspective of the New Testament eschatological statement of the Apostle Paul—"I consider that the sufferings of this present time are not worth comparing with the glory about to be revealed to us" (Rom. 8:28)—is in fact "an 'anti-theodicy' of sorts—an abandonment of all speculations to the problem of suffering" (Volf 1996, p. 138).

The Bible narrates acute suffering explicitly and implicitly. There are notable places where the subjects are explored, most famously in the Book of Job, where Job calls God to account for his actions and demands answers (e.g., Job 10, 31). Since no alternative theodicy is proffered, it could be argued that the Book of Job is also more of an anti-theodicy, rejecting the popular cause and effect theodic perspective presented by the friends' cycles of speeches and narrated in them being rebuked by God rather than Job (Job 42:7–9). In the New Testament, Jesus himself questioned God the Father as he hung on the cross—"My God, my God, why have you forsaken me?" (Matt. 27:46) using words from one of the many lament psalms, where other forms of interrogation are also used. Yet Jesus was content to commend his spirit into the hands of the Father even though no answer was forthcoming from God (Luke 23:46). Jesus rejected popular false theodicies (Luke 13:1–5; John 9:2–3), drawing on them only to re-orientate attention to more human practical and moral actions. The Apostles wrote of their own experiences of suffering and survival (2Cor. 11:23–29), ensuring that their readers recognised suffering and conflict as inevitable consequences of faithful discipleship to Christ (Phil. 1:29–30). One of the key joys of the Christian final solution, namely the eschaton, will be the abolition of all suffering and evil (Rev. 20:7–21:4). Therefore, the Bible is no stranger to issues and experiences of suffering and evil. In fact, biblical scholars are drawing our attention increasingly today towards the fact that much of the Bible has been written out of a context of catastrophic traumatic suffering (Carr 2014; Boase and Frechette 2016), where trauma provides a "powerful interpretive lens" (Boase and Frechette 2016, p. 1).

These issues of traumatic suffering in the Bible are not addressed philosophically or theoretically, to provide a theodicy. They are addressed theologically, but even then only up to a point. They are certainly addressed narratively, and this approach could well contribute towards a solution for the methodological problem I have identified.

Ethnography is recognised as a significant methodological approach for research into religious beliefs, with its ability to provide a more granular analysis of beliefs, constructed out of social narratives more than out of strictly religious imaginaries (Adeney-Risakota 2014; Taylor 2004), which can often differ, person to person (McGuire 2008; Spicknard et al. 2002; Hammersley and Atkinson 2007). In fact, we can narrow down the ethnographic method most suitable for our discussion to ethnographic theology. Ethnographic theology carries a loose hold on text-based theological normativity and universality. Traditional concepts of dogmatic or systematic theology centre around the notion of theological truths being absolute and normative in a universal sense. However, the ethnographer's discipline within the field of lived experience, affected by cultural, social, and even emotional factors, makes it evident that an individual's capacity to live by the principles of normativity inevitably fails, and the resultant reality of faith becomes very different to the normative ideal. Theologies and concomitant practices are constructed within cultural locales. Ethnographic theology is theology forged out of the realism of lived lives in the context of local culture, the habitus of the individual and of the community being significant. It does not abandon theological normativity altogether, but works collaboratively with it.

My principal research method, that of ethnography, focused upon capturing the narrative experiences of earthquake survivors and the bereaved from the Haiti earthquake. Amber Griffioen (2018), reflecting on the value of sufferer's narrative observes,

> In this sense, one's reflections on the divine nature and the role one assigns it in the story of one's suffering can bring meaning to one's cognitive, affective, and volitional chaos. It may also assist in promoting a kind of acceptance—not, perhaps, of the trauma or suffering one has undergone, but of the fact that one will always in some sense occupy a space that others lacking such a narrative will not.

In that sense, the lament Psalms have been so helpful to many sufferers. Helm's point, of course, is that sub-Christian ideas of that divine nature risk producing an element of falsity, or illusion into the story of one's own suffering. Therefore, theological as well as narratival integrity is required if, "the kind of character the Christian [sufferer-survivor] seeks to develop is a correlative of a narrative that trains the self to be sufficient to negotiate existence without illusion or deception" (Hauerwas 1981, p. 132). Such character development as Hauerwas recommends makes the value of the space afforded by academe important for the theological testing and weighting of such narratives.

Griffioen (2018, p. 8), while rejecting the speculative philosophical approach, argues for a more therapeutic approach, a small shift, in her view, in perspective to the question of suffering from the epistemic to the therapeutic, but still with a dynamic focus upon God. The *experience of the sufferer* is the precise location from which Griffioen suggests the most helpful data for guiding theological reflection can emerge. She comments, "I think the discipline of analytic philosophy of religion might do well to shift its attention somewhat from the theoretical God's-eye view to that of the existential and religious situations of those who really suffer—and from the epistemic status of religious belief to the practical situatedness of lived experience."

Once Griffioen's attention-shift is assumed, we are more likely to focus upon the more obvious human causal factors that make the practical situatedness of lived experience what it is, the factors that turn natural hazards into disasters. However, there is an aspect where I believe Griffioen's attention shift needs to be much larger than she does. Liberation theologians, working in low-income countries, have been filling the gap her perspective leaves for a long time (Griffin and Block 2013; Sobrino 2006; Gutiérrez 1984), with their insistence of attending to the systemic, structural societal factors.

Swinton (2007) makes a helpful distinction between theology as a mode of knowing that challenges cultural assumptions, and theology as a mode of knowing that is reactive to challenges that society asks. In particular, a practical theology that challenges cultural assumptions, when utilized in my context of understanding so-called natural disasters and in ways of response to such disasters, would identify the failures in human responsibility, which are more often structural and systemic, not simply individual and personal. The latter mode, reacting to challenges that society asks, is more likely to be asked by Western secular spectators who are asking questions of God that survivors rarely ask, and avoiding the human, structural, and systemic aspects that survivors identify most and are most concerned about. From the Christian tradition, the former theological approach becomes supremely practical, transformative, and helpful to the sufferers, the latter approach more intriguing for society's spectators on the "problem" of suffering. I concur with Griffioen (2018) when she complains that,

> the theodicies in mainstream analytic philosophy of religion stem from a place of relative privilege, in which the dominant voices represent those philosophers who are cognitively and emotionally in a position to be able to distance themselves from particular evils and traumata in a way sufficient to allow them to consider suffering more abstractly and to ask how it might be necessary for (or at least as conducive to) promoting some further divine end.

In making this judgment on the "privileged," Griffioen echoes Voltaire's protest against Leibnitz, mentioned earlier.

Such was the depth of suffering conveyed to me by my survivors, I would not even dream of considering any philosophical theodicy as an appropriate therapeutic resource for their anguish. To

have done so would risk causing more harm than good (Griffioen 2018; Kivistö and Pihlström 2017; Swinton 2007, pp. 3–4). However, it would be misleading to suggest that my explorations of philosophical theodicies in the academic laboratory have had no configurative influence upon my practical theological perspectives and practices.

Philosophers like Helm and Griffioen, liberation theologians like Gutiérrez and Sobrino, and practical theologians like Swinton and Ganzevoort, therefore, guide us towards a resolution of the conflicting methodological nexus mentioned earlier, and one that may reduce any irreconcilable polarisation between speculative theory and hard practice. Each of these scholars stresses suffering as a very practical human, existential experience.

Griffioen (2018) point about taking seriously the suffering person's testimony is also salient. As she states, "when we begin to take such testimony as a credible source of knowledge—when we trustingly listen rather than presumptively speaking—we may be moved to *resist* such evil and to stand together with and for those who suffer."

It was precisely for the reason she states—to take survivors' testimonies of their experience of the 2010 earthquake as "a credible source of knowledge"—that drove me to ethnographic fieldwork in Haiti, "to listen rather than presumptively speaking," with a view to exploring whether survivors' experience contributed toward a therapeutic narrative of recovery from their trauma.

## 4. Haiti Earthquake (2010)

From the Christian point of view, there are two important practical dimensions to catastrophic disasters that require addressing by any proposed solution: the ability of a survivor to recover as a human being in God's image, and both the individual and social structural learning for mitigating risk in the event of future hazards and incidents. Both dimensions, recovery and mitigation, are interconnected and should not be polarised.

However, when the focus is only upon the individual's coping and psychological recovery, while ignoring corporate systemic, structural factors, it creates a problem the Haitian, Brunine David, noted when he reflected upon the illegitimacy of spectator-commentators outside of Haiti claiming to speak for the Haitian survivors of the 2010 earthquake. His assessment was, "When they dare to talk about our courage and strength or perseverance, they change the meaning and take all the good from it and leave us with resilience; a kind of people who accept any unacceptable situation, people who can live anywhere in any bad condition that no-one else would actually accept" (Ulysse 2015, p. 61). In other words, such a focus can ignore the "unacceptable situation" and "the bad condition" that have been major contributors to the disasters, and which owe directly to human structural evils. (Griffin and Block 2013, pp. 1, 16, 55–70; Farmer 2008). My Haitian participants did not wish me to just tell the story of their sufferings, but also to tell the need for structural change in Haiti. Zarowsky (2004), analysing the dominant Western emphasis on the pathologically traumatised individual, as she reported her Somalian refugees' experiences of trauma, received a similar request:

> They did not wish me to stop at conveying their individual misery, for they knew it well enough and did not consider that emotional empathy was sufficient to resolving their difficulties ... If this insistence on building a politicized collective memory and master narrative challenging power and injustice from the local to the global represents "trauma," it is of a different scope and implies different therapeutic interventions than those suggested by conventional models of PTSD. [4]

Some kind of collaborative approach is necessary for addressing the two practical dimensions of personal, individual trauma and the structural causal factors in Haiti.

---

[4]	PTSD (Post Traumatic Stress Disorder) is a psychiatric disorder recognised in and described by the Diagnostic and Statistical Manual of Mental Disorders (DSM-IV). For further details, see (McFarlane and Girolamo, pp. 129–54) in (McFarlane and de Girolama 1996).

I conducted fieldwork in the worst earthquake-affected areas of Haiti in 2012–2015. My primary methodological approach to the project was ethnographic (in-depth interviews, observations and fieldwork journal) (Bennett et al. 2018; Kvale 2007). For this case study, I have in mind the particular existential data I gathered in the areas worst affected by the earthquake.

Participants self-identified as Christian (Catholic or Protestant), and they spoke of their faith being a helpful resource in recovering from their earthquake experience from a psycho-spiritual perspective.[5] In particular, the theology of the Bible, of creation, and of divine providence proved to be their most useful pastoral resources, especially the appeal to divine providence, as the following narrative from a woman with a long history of extreme poverty and suffering illustrates,

> It's still God who makes it possible for me to stay alive and gives me courage to keep on living. If I'm still living today it's he [God] who makes it possible for me to live. Even when everything becomes very dark for me, I know I've got God who will do everything for me. If I didn't, if I didn't believe in him, then, because of the twelfth of January [date of the earthquake], I wouldn't be here ... All the misery, all the poverty I'm going through at the moment, I leave it in God's hands. It's God who gives...that I'm in God's hands and at the disposal of God's will. I will always remain firm in my faith in regards to what happens, with regards to what I meet along the road.

In other words, she used her understanding of this doctrine as a hermeneutical tool for interpreting her pre-existing hardships and her experience of the earthquake. The result of adopting this theological perspective meant for her that God was in control of the earthquake. Other participants additionally saw the earthquake itself as a natural event, and a few even understood it involved the movement of tectonic plates. However, primarily in their view it was still an event under the control of a sovereign God, yet also fully compatible with human responsibility, as the following testimonies emphasise,

> I'd tell them it is a natural phenomena [sic], and as a natural phenomena anything can happen. I would not disagree with people that says that it was the will of God because everything that happens is within God's control. And so, but I mostly tell them that it is a naturally occurring disaster, because just like cyclones, cyclones have their natural [way of working].

> First and foremost, the earthquake is a natural phenomenon. It was all because here in Haiti we did not take the precautions to foresee such disasters. For me, that's my reason. My second reason I can say is that, in Haiti, we have three percent of vegetation. It is evident, having treated nature that way; it is evident that she would seek justice.

To the speculative, spectatorial mind, these views may leave wide open the question of why a good God would control nature in the form of an earthquake to bring about the deaths of over 200,000 people and the displacement of a million people (Daniell et al. 2011).[6] But such questions did not appear in the view of the great majority of participants, and they tell me that such questions did not appear in any negative form either in the minds of the people of faith they knew.

The overwhelming pastoral comfort participants gained was from the Bible and from the doctrine of divine providence. A recurring phrase participants used, repeated across the demographic spectrum, was "God doing his work" (Kreyòl: *Bondye ap fe travay. Li*). It struck me that this was a common descriptor for divine providence in the Haitian theological and cultural lexicon. It provided comfort and reassurance amid the chaos and suffering brought on by the earthquake: in spite of all that was

---

[5] All subjects gave their informed consent for inclusion before they participated in the study. The study was conducted in accordance with the Declaration of Helsinki, and the protocol was approved by the Ethics Committee of St. Edmund's College, University of Cambridge (Confirmation letter appended).

[6] The Haitian government gave 320,000, but suspicions were cast over this being an inflated figure in the political interest of gaining maximum foreign disaster aid. Many use the 220,000 figure.

going on, God, in sovereign providence, was going about his work. This phrase was never meant to lay blame at God's door, nor was it ever spoken out of terror, but it suggested a conviction that God is not beleaguered or diverted from his purposes; he *used* even an earthquake to do his work. This belief seemed to create a sense of normality, or of stability that brought reassurance to people in the midst of an event in which nothing else seemed to be normal or stable: not least, the ground they stood on and the buildings they lived in, solid, accustomed symbols of stability normally. It also provided a sobering perspective for Christian communities, confronting them with the fragility of life.

Though no one I interviewed used the term theodicy, the closest form of academic theodicy they did employ was that kind Swinton terms *practical theodicy*, as against the more speculative philosophical kind. Swinton (2007, p. 85) defines this kind of theodicy as

> the process wherein the church community, in and through its practices, offers subversive modes of resistance to evil and suffering experienced by the world. The goal of practical theodicy is, by practicing these gestures of redemption, to enable people to continue to love God in the face of evil and suffering and in so doing to prevent tragic suffering from becoming evil.

My participants' understanding of divine providence exemplified Swinton's focus upon practicing gestures of redemption inspired by the life, death, and resurrection of the Son of God (Swinton 2007, pp. 72–77; Abbott 2013, p. 251). Pastor John (not his real name) and his Church responded to the devastation and suffering from the earthquake in his town and outlying villages by practicing gestures of redemption inspired by the life, death, and resurrection of the Son of God. Immediately following the initial, devastating tremors, people from the town came to the destroyed church compound where they sang and prayed under the pastoral direction and care of Pastor John. The people remained within the compound for many weeks, fearing the insecurity within the town, and benefitting from the positive pastoral ambience created by Pastor John and the church. Here the church members shared their meagre food rations with the townsfolk. Pastor John and a Canadian colleague made vain ventures into the capital city, searching for food supplies, but only when the U.S. Marines landed near to the church compound, five days after the earthquake, were they able to obtain sufficient food and resources, which they shared among the people of Grand Goâve. Pastor John also visited his satellite churches, bringing food and resources to his members there for distribution around their villages. Thereafter, because of Pastor John's reputation in social justice activity in the town, people would come to seek advice and assistance from him rather than from the town mayor or from the civic authorities, whom they distrusted as dishonourable and corrupt.

It is the words of Forsyth (1916, p. 175) that perhaps most sum up the practical theodicy of my participants, when, in defence of his evangelical justification of God in the light of the horrors of the First World War, he wrote, "We do not see the answer; but we trust the Answerer, and measure by Him." The theological rationale for such trust, according to Forsyth, is the fact that "The only vindicator of God is God. And His own theodicy is in the cross of His Son Jesus Christ." The circumstances of the Haitian survivors routinely, let alone following a catastrophic earthquake, did not allow the luxury for philosophical speculation, nor did their experience of catastrophe seem to require it. Survivors simply leant heavily upon their faith in a God whose goodness was measured by his evangelical work in the cross and resurrection of Christ, in a wise and providential God who knew what he was doing, even though they did not understand all he was doing. What helped them most was their theology that worked relationally with God and with their suffering, not in isolation from, or in speculation upon, those relationships. When they lamented, it was in the context of their faith in God and love for him, not in contempt for him. They lamented *to* God as their friend, not *at* God as an enemy.

However, my research showed that theology was not the sole source of help the survivors made use of to cope with their earthquake experience. Another, equally helpful, resource was what I have called learned coping strategies, strategies passed down from generation to generation, as coping skills for the daily struggle most Haitians have always had with life in their country. The chic term for this amongst Western disaster specialists is "resilience." It is a term I resist using, in preference for learned

coping strategies. These strategies enabled my participants to resist succumbing to the pathologies of trauma, such as chronic forms of depression, grief, and PTSD.

I concluded, therefore, that the faith my Haitian participants had in divine providence and in Christ did provide a substantial therapeutic pastoral resource when it came to their coping with, and recovering from, their earthquake experience. However, whilst giving due regard to their theology working relationally in a positive way, what this theology did not do, by and large, was provide a compelling driver for participants to develop practical strategies for disaster risk reduction in the event of a future earthquake, even though the risk of another devastating earthquake occurring is an ever-present reality. (Calais 2015; Calais 2013; Frankel et al. 2010). Nor did their faith inspire any actions that sought to address the structural and systemic evils that make the Haitian population so vulnerable to seismic and meteorological hazards. These evils point most to issues of anthropodicy more than to theodicy. Forsyth (1916, p. vi) put the point well when he wrote, "The doubts that unsettle men most today are those that rise not from science but from society, not from the irrational but the unjust." Social justice issues are at the heart of what makes Haitians so vulnerable to the natural hazards that are part of her natural geology and geography.

## 5. Conclusions

I conclude that a practical theologized theodicy that incorporates elements of liberation theology can serve a therapeutic recovery more constructively than philosophical theodicy ever can. I make this suggestion because although many of my participants did not complain against God (albeit they did complain to God), they did complain against the State executives and against the elite class who connived in corrupt practices that maintained the majority population in poverty, which denied them access to justice, to education, and to basic social care (Farmer 2011, 2008, 2006; Hallward 2010; James 2010; Wilentz 1989, 2013). Liberation theology, adopted and adapted as a dynamic component of practical theology (Griffin and Block 2013; Chester 2005; Woodward and Pattison 2000; Gutiérrez 1984) is best suited to address the anthropodic challenges the natural hazard of the earthquake exposed so tellingly.

The reason so many people suffered and died, or were severely injured or displaced by the earthquake, was the widespread failure of buildings, to the collapse of infrastructure and the lack of disaster awareness. The empirical evidence for this conclusion is huge (Bankoff 2015; Bilham 2013; Ambraseys and Bilham 2011). However, the reasons for both the structural failure in buildings and for the lack of disaster awareness were not viewed by my participants as problems for a theodicy. They were seen as problems for an anthropodicy—challenges to the goodness of humans. The reasons so many suffered in the earthquake were entirely human, and they mostly boiled down to issues of social justice and poverty. My participants recognised the need for natural hazard education, but they often despaired at the lack of access to that kind of education, due to their poverty and to the way in which the systems worked. Access to effective education in Haiti is an economic, racial, class, and political issue. People have to pay for it, and if they are kept in poverty as they often are on the basis of colour or class (Farmer 2011, pp. 44, 52; Hallward 2010, p. 194), then they cannot afford to send their children to school, at least not beyond the primary level (Farmer 2011, p. 43; Final Report of the National Survey of Catholic Schools in Haiti 2012; Luzincourt and Gulbrandson 2010; Krebs 1971).

The requisite scientific information for understanding the seismic hazards is present in Haiti. It was there before the 2010 earthquake, and it is there even more so since (Mann et al. 2002, 1995, 1984; Manaker et al. 2008; Calais 2015). However, access to it in the public domain remains appallingly deficient and represents one of the most significant moral and social problems the country has to address before a future earthquake of similar proportions occurs. Without education, people cannot gain employment, and they cannot build up income capacity allowing them to build their homes safely, so they will die or get injured if another earthquake occurs. This was demonstrated most recently in October 2018 when a magnitude 5.9 earthquake happened in northern Haiti, killing 17 people, injuring

333, and displacing thousands. The Port-au-Prince daily newspaper (*Le Neuvelliste* 2018) reported, "The feeling of panic that seized every Haitian who felt the tremors, all over the country, and the deprivation of the institutions of Port-au-Prince as province showed that there is still work to reach the excellence in disaster preparedness like an earthquake." The scale of deaths, injuries, and damage to houses of this seismic event was out of proportion for an earthquake of this mid-range magnitude. Haitian earthquake specialist, Claude Prepetit, lamented, "If a magnitude 5.9 earthquake can do so much damage, imagine for a moment that the magnitude was the one we knew on 12 January 2010." (OCHA, UN Office for the Coordination of Humanitarian Affairs 2018; An Earthquake of Magnitude 5.9 Should Not Cause as Much Damage 2018).[7]

In addition to the complicating anthropodically challenging factors of poverty, education, employment, construction regulation, enforcement, and affordability, there is an added factor of corruption. Ambraseys and Bilham (2011) aver that corruption is endemic within the construction industry in the form of bribes to subvert inspection and licensing processes, as well as complicity in cost-cutting, quality-compromising practices. Death figures from earthquakes globally continue to rise alarmingly (Spence et al. 2011; Schlein 2010). Even so, death reduction from implementation of earthquake-resistant design can benefit earthquake prone countries, but only those "that have the wealth and willpower to mandate its use." Haiti does not have such wealth, nor the political willpower. Bilham (2013) lists three factors responsible for high death tolls from earthquakes and that also prevent the lessons of earthquake engineering being applied. These factors are: corruption in the building industry; the absence of earthquake education; and the prevalence of poverty. Again, each of these is relevant to the way Haiti suffered from the 2010 earthquake. Ambraseys and Ambraseys and Bilham (2011, p. 15) is sobering for Haiti: "The structural integrity of a building is no stronger than the social integrity of the builder, and each nation has a responsibility to its citizens to ensure adequate inspection." Social integrity in Haiti represents yet another structural problem to be addressed if lives are to be saved in the event of a future earthquake of even less-similar magnitude.

Among my participants, I found only a few for whom their theology provided, beyond lament, any motivation or direction when it came to addressing these deeply embedded structural evils. Though the human causal factors responsible for so much of their suffering as survivors of the earthquake and as survivors of life in Haiti generally are so empirical (Bankoff 2010; Smith 2006; Squires and Hartman 2006; Smith 2006; Oliver-Smith 2010; Oliver-Smith and Hoffman 1999), my participants often lacked any theology that motivated a protest for social justice. Practical theology, however, is accustomed to being interdisciplinary, especially with its collaborative trajectory with the natural and social sciences (Abbott 2013, pp. 38–40; Swinton and Mowat 2006, pp. 85–86, 255–258; Woodward and Pattison 2000), both of which are pertinent disciplines for natural disaster exploration and for making major contributions for disaster mitigation. My proposed practical theological/theodic methodological approach would also have the potential to take the pastoral effects of the doctrine of providence, that were so beneficial to survivors up to a point, to a stage beyond their current theological reach, namely, to combine with survivors' culturally learned strategies of survival to address the social and structural evils that belied the human causation of the earthquake disaster.

There are precedents for working for structural change in Haiti through religious communities. In the late 1970s, Fr. Jean-Paul Aristide, motivated by his liberation theology, saw the *ti kominote legliz* (the church community), and *ti legliz* (small church) movement emerge in Haiti. (Hallward 2010, pp. 15–16; Wilentz 1989, pp. 105–106). The local ecclesial communities became instrumental in organising protests seeking justice over the offences of the Duvalierist *Ton Ton Macout* (Aristide and Wilentz 1990). Unfortunately, such grassroots movements met with violent anti-Aristide repression by the military and elite classes in Haiti at that time, and there is no guarantee that a *ti*

---

7   The 2010 earthquake was magnitude 7.0, or thirty times the energy of the 2018 earthquake. I am grateful for this information given to me by Prof. Robert 'Bob' White, FRS., Professor of Geophysics in the Department of Earth Sciences at the University of Cambridge, U.K.

*legliz* today would not suffer similar reactions against them from the elites, given the currently volatile political climate and mood of the people. However, exploring a model for such grassroots faith-based actors who have a selfless passion for civic safety and who are sick of violence, conflict, and national humiliation in their nation that it is so vulnerable to, could become a model worth exploring and testing. What Haiti needs most is not the charity/aid-based solution it has laboured under for too long, since this only creates the need to perpetuate the aid industry (Thacker 2017, p. 205). According to our participants, Haiti requires profound change to eradicate the structural evil of poverty and the life-endangering evils associated with poverty.

From my participants' perspectives, Haiti requires "politicians with national agendas, not self-interest, one that recognizes its duty to its citizens." (Ulysse 2015, p. 8). Charity, and especially solidarity, have their place, but above all, "respecting the status of the poor as those who control their own destiny is an indispensable condition for genuine solidarity" (Griffin and Block 2013, p. 156). I agree with Gustavo Gutiérrez, therefore, that the solution for Haiti needs to be theological as well as political (Gutiérrez 1984, pp. 50–51). However, it will not be a philosophical theodicy that brings about the changes my participants long for, and suffer so much for want of. A practical theology of liberation, armed with a robust theology of human responsibility and accountability to God, would seem a more promising and effective practical theodicy than a philosophical theodicy that holds God accountable to humans.

A consuming focus upon philosophical theodicy has the potential as a massive distraction from the reality of life on the ground in Haiti after the earthquake, and this focus stands to divert attention away from the urgent human factors that have been proven to lie at the heart of why the natural hazard of the earthquake—so necessary to sustaining the enthralling earth we live in, and to the beautiful country Haiti can be—turned into a bearer of so much death, suffering, and destruction. When we can relay to the sufferer and to the bereaved from the earthquake what deeds survivors' and responders' faiths have carried for the survivors' relief and recovery, then we can come closer to fulfilling Hart's dictum: "... words we would not utter to ease another's grief we ought not to speak to satisfy our own sense of piety." In fact, actions will speak louder than any words, and for theology to be meaningful and authentic, it must produce actions, as the Apostle James informed us at the outset of this article.

**Funding:** This research was funded by The Templeton World Charity Foundation grant numbers TWCF0024 and TWCF0103 to The Faraday Trust for Science and Religion. The views expressed here are those of the author and not necessarily those of the funding agency.

**Acknowledgments:** I acknowledge the support given me by the Haitian survivor participants in sharing with me the worst moments in their lives. Without their willingness to do this, my research would not have been possible.

**Conflicts of Interest:** The author declares no conflict of interest.

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
