# Peer review of "“I Will Show You My Faith by My Works”: Addressing the Nexus between Philosophical Theodicy and Human Suffering and Loss in Contexts of ‘Natural’ Disaster"

_religions, doi:10.3390/rel10030213_

Round 1

Reviewer 1 Report

Lines 34-37: With regard to the ‘academic obsession for solving the mystery of suffering and evil, often in response to the theory of evolution’ – despite the several (presumably illuminating) quoted references, this point may benefit from an (even cursory) explanation.

Line 42 and then line 46 (also line 67 and 128): The author repeatedly calls him/herself an ‘academic theologian’ – although in the abstract he/she describes him/herself as ‘a researcher in the field of “natural” disasters, as well as being a disaster responder chaplain’. This seems a little confusing.

On Line 53, the correct year should be 1755.

Lines 62-66: While ‘obsessing’ about solving the big questions of theodicy is likely to remain an unsolved mystery, of little practical value to the sufferers of disasters, the reader would still benefit from a brief incursion into some of the questions, themes and perspectives that theodicy has had to offer.

Line 87: The line that says: ‘the polarity between the philosophical theodicy and the practical sort’ seems slightly unclear. Does the author mean the polarity between philosophical theodicy and practical approaches/theology?

Lines 128-129: Slightly repetitive – point has already been made in lines 42-43, 46-47, 67-68

Lines 217-220: Could it not be argued that ‘a theology of divine providence’ has itself an incipient  theodicean character – or that it is at least significantly related to a theodicean approach? The contrast between theodicy and divine providence could have been better explored.

Line 243: If the author prefers to use the Portuguese title and name, then the correct form would be ‘Marquês de Pombal’ – although it is unclear why the English version has been avoided (Marquis of Pombal).

Line 293: Perhaps this sentence could be rephrased to avoid the syntagm '(Christian) final solution’.

Line 351-353: The following sentence is very hard to follow/understand: ‘The latter mode is more likely to be asked by Western secular spectators to confront the challenges that believing in a good God pose in an ideological preference for not believing in such a Being.’ I would suggest rephrasing.

364: Line should end with a full stop.

Lines 372-373: Now would have been – perhaps - a good opportunity to reformulate or recap what this ‘resolution of the conflicting methodological nexus’ is (for more clarity).

Lines 452-453: The author earlier wrote that ‘less than a handful [of participants] raised any desire to interrogate God or to hold God to account in a negative way for their plight.’ It is not entirely convincing that holding God to account ‘in a negative way’ is regarded as a form of theodicy, while holding him to account ‘positively’ seems to be related to ‘divine providence’. The predominant (‘positive’) view among the Haiti sufferers referring to ‘God doing his work’ does seem like an attempt to theodicise.  

The premise of this article seems to be to contrast – when dealing with difficult pastoral cases following natural calamities – a passive, philosophical, reflective theodicy, with a ‘practical theodicy’, which would attempt a more constructive and healing approach, while also paying attention to the human factor of causality and presumably also facilitating a social and structural change/development. While the former (theoretical/philosophical) approach’s lack of usefulness is rather self-evident and the study could have saved time and energy in trying to demonstrate its deficiency, the latter approach (of a ‘practical’ theodicy – or indeed theology) would have benefited from a more in-depth treatment.

Author Response

Dear Reviewer,

 Thank you for your helpful, constructive comments on the article. I attach my responses, and revisions can be seen in a revised version of the article with the Tracking turned on. I hope you find these satisfactory. I think they give strength and clarity to the revised article. 

Reviewer 2 Report

Masterful - restores the anthropocentric focus on the origin of sin and suffering in the biblical narrative - in a sound and healthy theological manner.  I hope you are considering a book length project! This will be required reading in sections of my courses dealing with theodicy.   

Author Response

Dear Reviewer,

 Many thanks for your positive review and encouraging words of comment. I do have plans to take exploration of the theme further into a book project. I hope your students will benefit from the article if it is published, and from any further publications from me on this important theme.

Reviewer 3 Report

I thought the article was excellent and I hope it is published. I used to teach a course on the Holocaust and I would have liked to use an article like this in presenting the issue of theodicy in its more practical-theological form.

Author Response

Dear Reviewer,

 Man thanks for your approval and encouraging comments on the article. I do hope to be able to publish this article and also explore the topic further into a book.